# Leveraging Fully Observable Policies for Learning under Partial Observability

**Hai Nguyen, Andrea Baisero, Dian Wang, Christopher Amato, Robert Platt**
Khoury College of Computer Sciences
Northeastern University, Boston, MA, United States
`nguyen.hai1@northeastern.edu`

**Abstract:** Reinforcement learning in partially observable domains is challenging due to the lack of observable state information. Thankfully, learning offline in a simulator with such state information is often possible. In particular, we propose a method for partially observable reinforcement learning that uses a fully observable policy (which we call a *state expert*) during offline training to improve online performance. Based on Soft Actor-Critic (SAC), our agent balances performing actions similar to the state expert and getting high returns under partial observability. Our approach can leverage the fully-observable policy for exploration and parts of the domain that are fully observable while still being able to learn under partial observability. On six robotics domains, our method outperforms pure imitation, pure reinforcement learning, the sequential or parallel combination of both types, and a recent state-of-the-art method in the same setting. A successful policy transfer to a physical robot in a manipulation task from pixels shows our approach's practicality in learning interesting policies under partial observability.

**Keywords:** Partial Observability, Imitation Learning, Fully Observable Experts

## 1 Introduction

Many real-world robotics control problems feature some degree of partial observability, but policy learning in this setting remains a significant challenge in robotics [1]. While there are many reinforcement learning methods that address partial observability in theory [2, 3, 4, 5], their performance remains questionable in practice in interesting partially observable (PO) domains which require long-term memorization and active information gathering [6, 7]. In contrast, the setting of fully observable (FO) control has featured the success of many powerful reinforcement learning (RL) algorithms (*e.g.*, [8, 9, 10, 11]). Unfortunately, full observability only holds for a small portion of realistic robotics problems.

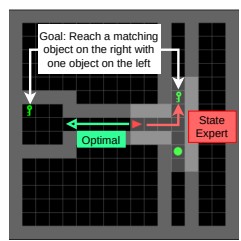

Figure 1: To reach the correct goal object, a state expert takes the red path directly, while a partially observable agent must first take the green path to identify the correct goal object, then take the red path.

In this work, we attempt to leverage good *fully observable* policies (*state experts*) available only during offline training to help train PO policies that can execute online. We rely on the setting of *offline training and online execution*, a successful RL framework where an agent can use "privileged" information such as the state [12, 13, 14, 15] or the belief about the state [6] during offline training, *e.g.,* from simulators, to efficiently learn PO policies that are later can be deployed without the access to the privileged information anymore. In this work, the privileged information is not just the state itself but also the state expert. Our setting can be illustrated in a navigation task (Figure 1), which requires an agent to navigate to an unknown goal object on the right, identifiable by an object on the left side. While the optimal behavior under partial observability is to first navigate leftwards to identify the goal object, the state expert is able to move to the goal object directly. Despite being sup-optimal from the PO perspective, the state expert can provide experience during training leading to the goal object, which is potentially useful for both exploration and as a part of the policy needed in the PO case after the goal object is identified.

6th Conference on Robot Learning (CoRL 2022), Auckland, New Zealand.

In this work, we propose to perform a form of *cross-observability soft imitation learning* (COSIL), *i.e.*, *softly* projecting the behavior of the FO agent into a PO agent which still tries to maximize its own performance. FO experts are generally easier to obtain than PO ones, which often require more impractical computations, *e.g.,* some sufficient statistics of the entire history like the belief states, which requires the true environment dynamics [16]. The resulting COSIL agent balances behaviors that imitate the FO agent with behaviors that are necessary for PO optimality. Experimental results on six robotics domains show that our proposed COSIL method performs significantly better than pure imitation learning, pure reinforcement learning, the sequential or parallel combination of the two types of learning, and other state-of-the-art methods in the same setting. Moreover, a policy with pixel observations learned in simulation is transferred successfully to a physical robot, showing COSIL's ability to learn policies that can handle the complexities of the real world. Additional details can be seen at our project website `https://sites.google.com/view/cosil-corl22`.

## 2   Related Work

Imitation learning methods such as behavioral cloning (BC) [17] and DAgger [18] perform policy learning by minimizing a supervised loss between the expert's actions and the agent's. These agents are trained on a dataset of encountered states and expert actions and are utilized successfully in tasks like autonomous flight [19, 20] or self-driving [21]. Several methods combine reinforcement learning and imitation learning (often from limited expert demonstrations), *e.g.*, DQfD [22], DDPGfD [23], and GAIL [24]. However, these methods are designed for Markov decision processes (MDP), *i.e.*, both the expert and the student can observe the environment state. Closest to our work are ADVISOR [25] and Asymmetric DAgger (A2D) [15], which leverage FO experts *during training* for PO tasks. A2D operates in a setting requiring a differentiable and modifiable state expert, jointly trained from scratch (using the environment states) with the student agent, while *we assume a given, fixed state expert*. In the same setting as ours, ADVISOR adaptively changes the weighting per state between the imitation loss and the reinforcement learning loss (see the appendix for a brief description of ADVISOR). However, it also requires training an additional PO actor to mimic the state expert to compute the weighting. Therefore, both ADVISOR and A2D must wait until the additional actor (ADVISOR) and the state expert (A2D) are properly trained before utilizing them to better train the main agent. In contrast, our method can utilize the state expert immediately.

## 3   Background

In this section, we introduce the background topics required to understand our contributions, *i.e.*, partially observable Markov decision processes, and the Soft Actor-Critic algorithm.

### 3.1   Partially Observable Markov Decision Processes

A partially observable Markov decision process (POMDP) [26] is defined by a tuple $(\mathcal{S}, \mathcal{A}, \Omega, p_0, T, R, O, \gamma)$, where $\mathcal{S}$, $\mathcal{A}$, and $\Omega$ are the state space, the action space, and the observation space, respectively. The initial state is sampled according to the starting state distribution $p_0(s)$, and the next states are sampled according to the stochastic transition function $\Pr(s' \mid s, a) = T(s, a, s')$. The agent does not observe the states but rather receives observations sampled from the observation function $\Pr(o \mid a, s') = O(s', a, o)$. In order to act optimally, a PO agent must, in general, condition its actions on the entire action-observation history $h_t = (a_{<t}, o_{\leq t})$, *i.e.*, all the observable information it has seen so far [27]. Denoting the space of histories as $\mathcal{H}$, the goal is to find a *history* policy $\pi \colon \mathcal{H} \to \Delta(\mathcal{A})$ which maximizes the expected $\gamma$-discounted return $J_\pi = \mathbb{E}\left[\sum_{t=0}^{\infty} \gamma^t R(s_t, a_t)\right]$.

Although our work focuses on PO control, it also involves FO agents modeled as a *state* policy $\mu \colon \mathcal{S} \to \Delta(\mathcal{A})$. To differentiate them clearly, we denote history policies as $\pi$ and state policies as $\mu$.

### 3.2   Soft Actor-Critic

Soft Actor-Critic (SAC) [10, 28] addresses the FO maximum-entropy control problem, *i.e.*, the problem of finding a policy $\mu$ which solves a given FO control problem while maintaining a high action-entropy when possible. SAC does this by extending the standard RL objective with a weighted

entropy term,

$$J_\mu = \mathbb{E}\left[\sum_{t=0}^{\infty} \gamma^t \left(R(s_t, a_t) + \alpha H(\mu(s_t))\right)\right] , \tag{1}$$

where $H(\mu(s)) = \mathbb{E}_{a\sim\mu(s)}\left[-\log\mu(a\mid s)\right]$ is the entropy of policy $\mu$ at the given state $s$, and $\alpha > 0$ is a trade-off coefficient which determines the relative importance between the RL objective and the entropy-maximization objective. The entropy term can be interpreted as a supplementary pseudo-reward given to the agent, which is high for high-entropy policies and low for low-entropy policies. Therefore, the agent will seek not only to maximize the entropy of the policy in the visited states but also to visit states associated with a high-entropy policy.

In practice, SAC is an off-policy learning algorithm that employs a replay buffer containing past transitions $\mathcal{D} = \{(s, a, r, s')_i\}_{i=0}^{N}$. SAC trains a parametric policy model $\mu_\theta\colon \mathcal{S} \to \Delta\mathcal{A}$, and a parametric value model $Q_\phi\colon \mathcal{S} \times \mathcal{A} \to \mathbb{R}$. The policy model is trained by maximizing

$$J_\mu(\theta) = \mathbb{E}_{s\sim\mathcal{D}, a\sim\mu_\theta}\left[Q_\phi(s, a) - \alpha\log\mu_\theta(a\mid s)\right] , \tag{2}$$

while the value model is trained to minimize

$$J_Q(\phi) = \frac{1}{2}\mathbb{E}_{s,a\sim\mathcal{D}}\left[\left(R(s, a) + \gamma\mathbb{E}_{s'\mid s,a}\left[V(s')\right] - Q_\phi(s, a)\right)^2\right] , \tag{3}$$

$$V(s) = \mathbb{E}_{a\sim\mu_\theta(s)}\left[Q_{\bar\phi}(s, a) - \alpha\log\mu_\theta(a\mid s)\right] , \tag{4}$$

where $Q_{\bar\phi}$ is a frozen target model that is updated at a slower pace than $Q_\phi$ to improve stability.

Hyperparameter $\alpha$ plays a central role in SAC, determining how much high-entropy states are preferred to pure rewards. Choosing a reasonable $\alpha$ can be difficult since it is not directly interpretable, and a good value depends dynamically on the current policy's expected returns and entropy. Haarnoja et al. [28] proposed to automatically adjust $\alpha$ by minimizing its own objective,

$$J_\alpha(\alpha) = \alpha\mathbb{E}_{s\sim\mathcal{D}, a\sim\mu_\theta(s)}\left[-\log\mu_\theta(s)\right] - \alpha\bar H , \tag{5}$$

where $\bar H$ is a given target entropy. In practice, Equation (5) modulates $\alpha$ such that it is increased if the current policy entropy is lower than the target entropy and vice versa. In contrast to choosing a value of $\alpha$, choosing a value of $\bar H$ is much simpler since it is broadly interpretable as the logarithm of the number of actions that we want the max-entropy policy to consider in an average state.

### 3.3 Soft Actor-Critic for Partially Observable Control

Although primarily designed for FO control problems and state policies $\mu$, SAC can be easily adapted to handle PO control problems and history policies $\pi$, like most (if not all) model-free learning algorithms. Two main changes need to take place for SAC: First, all appearances of a state $s$ in the equations and models of SAC must be replaced with a respective history $h$. This also implies the use of a recurrent neural network component in the overall architecture of policy and value models, *e.g.*, an LSTM [29] or a GRU [30]. Second, the replay buffer must be structured in order to contain and extract (truncated) episodes rather than individual transitions.

## 4 Cross-Observability Soft Imitation Learning

SAC is based on the premise of max-entropy control and is designed to find policies that solve the control task while pertaining to high entropy. This is achieved by extending the standard RL objective with an entropy component which not only pushes the policy model to be more stochastic for any given state but also acts as a pseudo-reward that pushes the policy to visit future states where the policy can be more stochastic. Inspired by this dual effect, we aim to employ a similar technique to exploit FO expert knowledge to train a PO agent.

Consider an offline training scenario where an FO expert $\mu$ is available, *e.g.*, obtained via a planning procedure, a pre-trained model, or a model trained jointly with the PO agent. We propose formulating a pseudo-reward for the PO agent $\pi$ based on the expected similarity with the FO agent $\mu$, expressed as the following *cross-observability soft imitation learning* (COSIL) objective,

$$J_\pi = \mathbb{E}\left[\sum_{t=0}^{\infty} \gamma^t \left(R(s_t, a_t) - \alpha D(\mu(s_t), \pi(h_t))\right)\right] , \tag{6}$$

where $D(\mu(s), \pi(h))$ is some divergence measure between the action-distributions of $\mu$ and $\pi$, *e.g.*, the *KL* divergence $D(\mu(s), \pi(h)) = \mathbb{E}_{a\sim\mu(s)}[\log\mu(a \mid s) - \log\pi(a \mid h)]$, or the *total variation* divergence $D(\mu(s), \pi(h)) = \max_a |\mu(a|s) - \pi(a|h)|$, among other options.

Minimizing the divergence $D(\mu(s), \pi(h))$ can be interpreted as a form of *cross-observability* imitation learning, which tries to project FO behaviors as PO behaviors. On its own, such a form of imitation learning is known to be sub-optimal, *e.g.,* it is not generally possible to replicate the behavior of $\mu$ with $\pi$. Further, FO behaviors are known to be, in general, sub-optimal for PO control, *e.g.*, PO agents might need to engage in information-gathering behaviors which an FO agent would not exhibit. An agent $\pi$ which strictly minimizes the divergence $D$ would incur an *optimality gap* between its resulting performance and the true optimal performance (see the appendix).

Despite these issues, we argue that applying a trade-off between the pure RL objective and the imitation learning objective in a *soft* fashion, *i.e.*, using the divergence $D$ as a pseudo-reward can be quite beneficial. We are directly inspired by SAC, which applies a similar trade-off between the pure RL and the max-entropy objectives to establish itself as a state-of-the-art model-free RL algorithm. In SAC too, the max-entropy objective alone is insufficient and inadequate when considered alone and only becomes beneficial when combined with task rewards as a pseudo-reward. Additionally, the FO expert $\mu$ is likely to be a good source of exploratory behavior even for the PO task. The actions chosen by $\mu$ are undoubtedly related to the overall task, and it is likely beneficial for the PO agent to explore them thoroughly. Second, there are many PO tasks where the agent can and/or must reach low state uncertainty to solve the tasks. Under such low state uncertainty situations, the optimal PO and FO behaviors overlap strongly, if not perfectly (as long as the state uncertainty is maintained low enough). In such situations, the optimality gap becomes low, and the cross-observability imitation learning task becomes beneficial. Finally, because the soft-imitation task is encoded as a pseudo-reward, the PO agent is generally pushed to achieve situations where there is low state uncertainty, which tends to be very beneficial for PO agents.

Similar to Equation (5), we formulate a minimization objective for $\alpha$,

$$J_\alpha(\alpha) = \alpha\bar{D} - \alpha\,\mathbb{E}_{h,s\sim\mathcal{D}}\left[D(\mu(s), \pi(h))\right],\tag{7}$$

which modulates $\alpha$ dynamically in order to maintain an expected divergence of $\bar{D}$. Like the target entropy $\bar{H}$ in SAC, $\bar{D}$ is an important hyperparameter that indicates how different the policies $\pi$ and $\mu$ are allowed to be on average. The semantics of $\bar{D}$ depend on the choice of divergence function $D(\pi(h), \mu(s))$, and good values of $\bar{D}$ are likely to be domain-dependent. In practice, we perform a grid search to find the best value. Please see the appendix for the detailed algorithm.

## 5  Experiments

We perform experiments on a diverse set of robot and navigation domains with discrete (**D**) and continuous action spaces (**C**), proprioceptive and pixel observations. We focus on the setting where the state experts are given *during training*, *i.e.,* the setting of ADVISOR [25].

### 5.1  Domains

Below, we only briefly describe the domains and the behavior of the state experts; see the appendix for more details.

**(D) Bumps-1D.** [6] Two movable homogeneous bumps rest on a table, with a robotics finger moving horizontally above them (Figure 2a). When an episode starts, the positions of all entities are randomized, but the left-right order between the bumps is maintained. The agent must push the rightmost bump (blue) to the right without disturbing the left bump (red). There are four action combinations: move left or right, each with a compliant or stiff finger. The agent does not know which bump is rightmost – it must infer using the history of the finger's deflected angles and positions while touching both bumps with a compliant finger before pushing right the blue bump with a stiff finger.

**(D) Bumps-2D.** [6] The same finger is now constrained to move in a plane above two bumps of different sizes (Figure 2b). It must reach the bigger bump (red) and stay there to get the only non-zero reward. However, it is unaware of which bump is bigger and, therefore, must make contact with

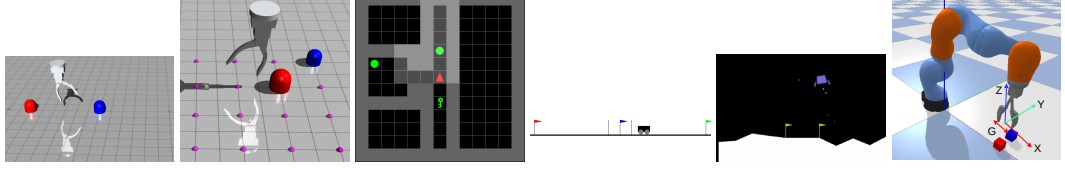

(a) Bumps-1D  (b) Bumps-2D  (c) Mini-Memory  (d) Car-Flag  (e) LunarLander  (f) Block-Picking

Figure 2: Our experimental domains. The first three domains have discrete actions and the last three feature continuous actions. `Block-Picking` is the only domain with pixel observations.

both bumps at least once, inferring the bumps' relative sizes from the angular displacement of the finger. The state expert, however, knows the position of the bigger bump and can go there directly.

**(D) Minigrid-Memory.** [31] An agent (red triangle) must go to a matching object located on the right side with an object (a ball or a key) located in a small room on the left side (Figure 2c). The grid layout, the matching object type, and the agent's starting position are randomized per episode. The agent might start in a cell where it cannot see the object; therefore, an optimal agent must first gather information by going/turning left into the small room to see the object. We chose the biggest version of this environment [31] with a size of 17.

**(C) Car-Flag.** [32] A car must reach the green flag, which can be on either the left or right side. The agent normally can only observe the car's position and velocity, but if it is near the blue flag, it can also observe the green flag's side (left/right). The optimal policy will be navigating to the green flag *only* after visiting the blue flag to find the right side. In contrast, the optimal state expert will always go left or right towards the green flag, whose position is known to the expert.

**(C) LunarLander-P, LunarLander-V.** We consider two versions of the classic LunarLander environment [33] where the agent only observes subsets of the full state (see the appendix for more details). During training, the expert, on the other hand, can observe the full state. Masking parts of the state to turn MDPs into POMDPs is common in previous work [34, 3, 35, 5]. However, these domains do not explicitly require active information gathering because the missing information (*e.g.*, velocities) can be inferred only by memorizing short observation histories (*e.g.*, positions).

**(C) Block-Picking.** Two homogeneous blocks rest on a table. The red block is fixed to the table and cannot be moved. In contrast, the blue one is movable, and the agent will accomplish the task if it picks the blue block. The agent takes in a top-down colorless depth image; therefore, it must gather information by trying to move both blocks to check for movability. The state expert knows the poses of the movable block during training and can always perform a successful pick. This task is particularly challenging if learning from scratch because the exploration is hard, the reward is sparse, the action space is continuous, and it can potentially be long-horizon.

## 5.2 Baselines

We compare COSIL against a diverse set of pure imitation learning agents, pure reinforcement learning agents (general and specialized for POMDPs, on-policy, and off-policy), and the combination of both types in several ways. If not described explicitly, all of these agents are *memory-based*.

**ADVISOR.** ADV-On is the original on-policy ADVISOR [25] that is built off PPO [9]. For a fair comparison with COSIL (off-policy), we implemented an off-policy version of ADVISOR (ADV-Off) using SAC for continuous action spaces [10, 28] and discrete action spaces [36].

**DAgger** [18] is chosen to represent imitation learning methods instead of a naive behavioral cloning agent, because DAgger often performs better.

**TD3** refers to a recurrent version of TD3 [37] for domains with continuous action spaces, implemented in [34].

**SAC** refers to two recurrent versions of SAC for continuous actions [10, 28] and for discrete actions [36], followed the implementation by [34].

**BC2SAC** is pre-trained with behavioral cloning for $p_{BC}\%$ of the total training timesteps before being trained with SAC.

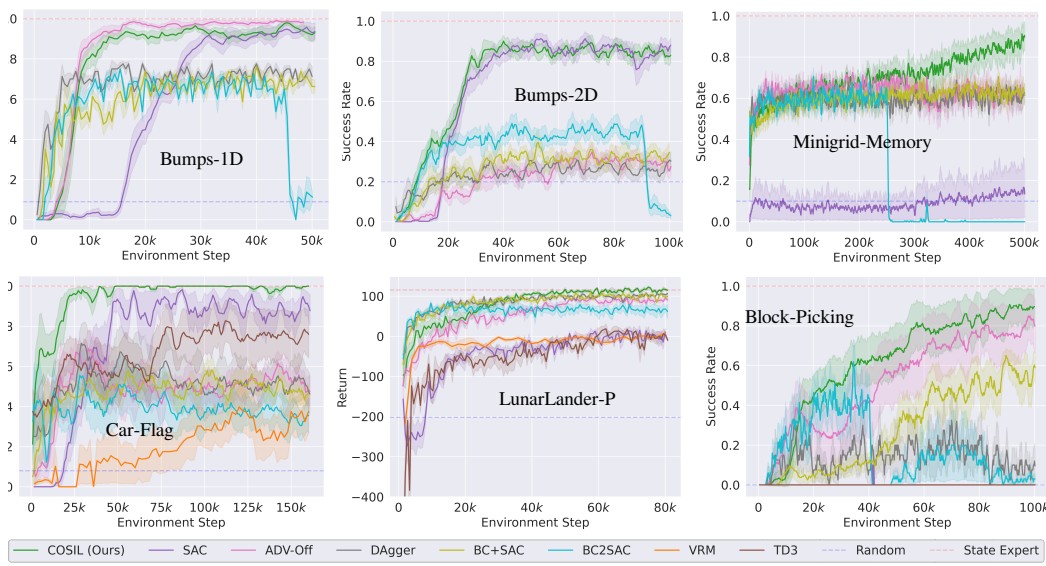

Figure 3: Learning curves of all methods (4 seeds with the shaded areas denoting 1 standard deviation). Note that all agents (except Random and State Expert) are memory-based agents.

**BC+SAC** is trained using a weighted combination of behavioral cloning loss $\mathcal{L}_{\text{BC}}$ and RL losses $\mathcal{L}_{\text{RL}}$, *i.e.*, the agent is trained by optimizing $\mathcal{L} = w_{\text{BC}}\mathcal{L}_{\text{BC}} + (1 - w_{\text{BC}})\mathcal{L}_{\text{RL}}$.

**VRM** is a state-of-the-art off-policy model-based agent [3] for POMDPs with continuous action spaces by combining a recurrent variational dynamic model and a SAC agent.

**Random, State Expert.** While a random agent takes actions randomly, a state expert is either trained with full state access or is hand-coded using the domain knowledge.

**Additional Baselines** include recurrent **PPO** [9] implemented in [38], an on-policy POMDP specialized method - **DPFRL** [2], and a non-recurrent SAC (**Markovian SAC** [34]) that uses observations instead of histories. We report the performance of these baselines in the appendix.

### 5.3 Experiments

**Metrics.** Depending on each domain, we record either the success rates or the returns of evaluation agents averaged over ten episodes. For SAC-based agents, we turn off exploration during evaluation. The reported results are averages over four seeds with shaded areas denoting one standard deviation with optional smoothing if needed. For Random and State Expert, their performances are averaged over 100 episodes and visualized as horizontal lines.

**Results.** The performance of agents is shown in Figure 3 (the performance in `LunarLander-V` is similar to `LunarLander-P` and is reported in the appendix). Overall, COSIL (green) is comparable to ADV-Off in `Bumps-1D`, `LunarLander-P`, `Block-Picking` but considerably better in the remaining three domains, which we argue have larger optimality gaps. DAgger performs quite well in `LunarLander-P`, suggesting that this domain's optimality gap is small. This is expected for this domain because the partially observable angular velocities of the vehicle can be inferred from a few recent fully observable angles. Our agent can also perform well in this domain, signifying the ability to work in domains with a small optimality gap. Other baselines that do not use the state expert struggle to learn efficiently, especially in `Block-Picking`, where random exploration will have a minuscule chance of picking the correct object. This explains the complete failures of naive baselines such as SAC or TD3, and highlights the advantage of exploiting the state expert for better exploration. BC2SAC performs poorly across all domains. This way, reinforcement learning often unlearns imitation learning policies when optimizing RL losses. The poor performance is illustrated by sudden drops whenever reinforcement learning starts. BC+SAC does not seem to perform well either, often not significantly better than DAgger. A fixed static combination like BC+SAC can be sub-optimal because the optimal PO policy can sometimes act vastly differently from the FO expert.

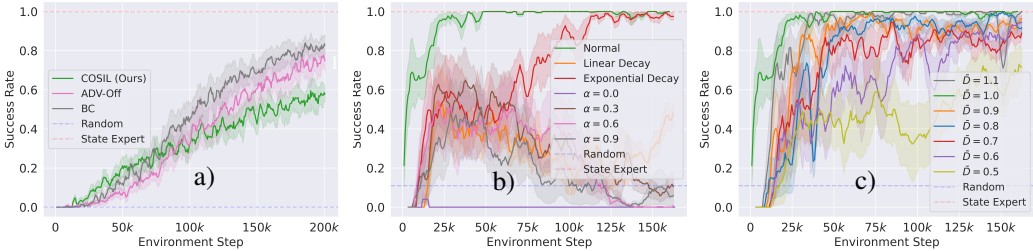

Figure 4: Additional results: a) COSIL in `MiniGrid-Crossing`; Performance of COSIL in `Car-Flag` when: b) linearly/exponentially decaying $\alpha$, varying $\alpha \in \{0.0, 0.3, 0.6.0.9\}$; c) varying $\bar{D} \in \{0.5, 0.6, 0.7, 0.8, 0.9, 1.0, 1.1\}$.

For instance, the PO agent often must act to gather more information about the state. In contrast, the state expert hardly has to do that, given its full knowledge of the state. Therefore, it is difficult for BC+SAC to behave optimally by fixing the weights between BC and RL loss. In fact, this observation has indeed motivated ADVISOR, which adaptively changes the weights between the BC loss and the PPO [9] loss for each state (see the appendix for more details).

**Training & Hyperparameter Tuning.** All agents are trained using complete episodes (truncated if too long). On-policy agents (*i.e.*, DPFRL, PPO, and ADV-On) use 5-10x more samples than other off-policy agents to train. We perform a grid search over relevant hyperparameters of each method (see the appendix for more details), then select the best combinations to report.

## 5.4 Additional Experiments

**Performance in Domains with Small Optimality Gap.** When the optimality gap is small, *e.g.*, when the state expert is nearly optimal under both full and partial observability, the best learning strategy is to purely mimic the state expert. We demonstrate this point in a variant of `MiniGrid-Crossing` [31] with a grid size of 25 and 10 crossings (see the appendix for details about the domain). This domain has a very small optimality gap: the paths that lead to a goal cell of an optimal PO agent (which only observes a local area around it) and a state agent (which observes the whole grid world) are quite similar. Figure 4a shows that both COSIL (with $\bar{D} = 0$) and ADV-Off are outperformed by behavioral cloning as expected. In the case of COSIL, by setting $\bar{D} = 0$, $\alpha$ will be increased to penalize the agent heavily if it behaves differently from the state expert (see Equation (6)), gradually morphing COSIL into behaving like BC.

**Not adapting $\alpha$ over time.** In Figure 4b, we illustrate the performance of COSIL in `Car-Flag` without adapting $\alpha$ to reach the target divergence $\bar{D}$. We investigated reducing $\alpha : 1 \rightarrow 0$ linearly/exponentially over time or keeping it fixed at different values in $\{0.0, 0.3, 0.6, 0.9\}$. The figure shows that adapting $\alpha$ is superior to other ways of modulating $\alpha$ in this domain. However, similar to SAC, there is no hard constraint that $\alpha$ must be always adapted.

**Performance with varying $\bar{D}$.** We investigate the sensitivity of COSIL in `Car-Flag` when varying $\bar{D}$ from 1.1 to 0.5 with a step of 0.1. Figure 4c shows COSIL is stable with half of the tested range.

## 5.5 Real Robot Experiments

We transfer the policies learned in simulation in `Block-Picking` to a Universal Robot UR5 arm mounted with a Robotiq 2F-85 gripper (Figure 5). We use two cameras: one Occipital Structure Sensor (Cam 1) and one Microsoft Azure Kinect camera (Cam 2), and combine their point clouds to create a depth image by using an orthographic projection at the gripper's position. To better transfer, during training, we add uniform noise to the positions and the yaw angle of the two blocks and add Perlin noise [39, 40] to the simulated depth images. We pick the best policies in the simulation of ADV-Off and COSIL, roll out ten episodes for each, and count the number of successful picks.

During the deployment, when the agent manipulates the supposedly immovable block, we temporarily disabled the movements along $x, y, g$ axes (see coordinates in Figure 2f) so that the agent will manipulate the other block after a while trying to move the earlier block without observing any

movements. The disabling is necessary because when the agent observes that the immovable block is *movable*, it will persistently manipulate that block. The temporal blocking allows us to avoid heavily gluing the immovable block to the table. However, it also leads to cases that the agents never see during training (the agents desire to move the gripper, but nothing moves due to the blocking), but both COSIL and ADV-Off seem to generalize well.

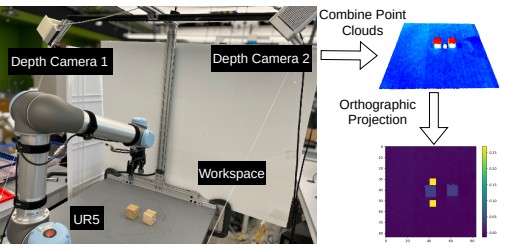

Figure 5: Experimental robot setup.

| Method | Simulation | Real-world |
|--------|------------|------------|
| ADV-Off | 9/10 | 6/10 |
| COSIL | 9/10 | 8/10 |

Table 1: Result of transferring policies learned in simulation to the real world in `Block-Picking`.

Table 1 shows the performance comparison of pick successes between COSIL and ADV-Off. The results show that our COSIL policy learned in simulation can actually perform well in the real world, in a PO task that would be very challenging to learn without the aid from a state expert during training. Empirically, our policies are more robust to rotational and translational misalignment of the two blocks (see videos on our project website), leading to more successful picks. However, similar success rates are expected with a bigger sample size, given their similar performance in simulation.

## 6 Limitations

COSIL can bias exploration negatively if the state expert is heavily sub-optimal under partial observability, being equal to giving very poor demonstrations. In this case, it is doubtful that any method can utilize FO experts to aid in learning optimal PO policies. To illustrate, we modify `Bumps-2D` such that staying on top of the bigger bump without visiting the smaller bump in the past will be penalized heavily (-100). Previously, such behavior of the state expert did not incur any penalty. Moreover, to maintain Markovian rewards (*e.g.*, the reward only depends on the last state and action), we add a flag indicating whether the smaller bump was visited to the observation. Figure 6 shows that both COSIL and ADV-Off are outperformed by SAC, with ADV-Off failing to learn completely. For COSIL, setting a high value of $\bar{D}$ might alleviate the situation as COSIL will become closer to a normal SAC agent.

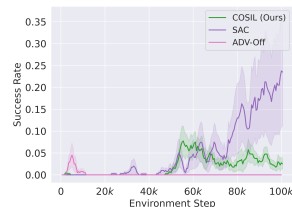

Figure 6: An example when COSIL can be less performant than SAC.

Another limitation is the requirement of constant access to expert actions at every state during offline training, which might be impractical. One potential fix is using our method with a variant of SAC that learns from limited demonstrations, such as one proposed by Liu et al. [41]. Another possible direction for addressing this issue is to use a method proposed by Gangwani et al. [35] with limited demonstration trajectories given by a state expert instead of a POMDP expert.

## 7 Conclusion

This paper tackles POMDPs in robotics from an unconventional angle: leveraging fully observable policies during offline training to better train partially observable policies that later can be deployed online. We introduce *cross-observability soft imitation learning*, which balances achieving high PO performance with acting similarly to the FO expert. With COSIL, the PO agent can exploit the FO expert for targeted exploration and information-gathering during offline training. COSIL outperforms the state-of-the-art method in the same setting in several robotics domains. COSIL also performs better than other conventional approaches and can learn policies that are successfully transferred to a physical robot.

**Acknowledgments**

This material is supported by the Army Research Office under award number W911NF20-1-0265; the U.S. Office of Naval Research under award number N00014-19-1-2131; NSF grants 1816382, 1830425, 1724257, and 1724191.

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
