# OpenReview forum: "Leveraging Fully Observable Policies for Learning under Partial Observability"
_robot-learning.org/CoRL/2022/Conference — CoRL 2022 Poster_

### Official Review · Reviewer_a2e5 · 2022-07-29

**Originality:** Good
**Technical Quality:** Very Good
**Clarity Of Presentation:** Very Good
**Impact:** 2

**Recommendation:**

Weak Accept: I recommend accepting the paper, but will not argue for my recommendation if the majority of other reviewers have a different opinion.

**Summary:**

This work studies an imitation learning setting in which the learner is subjected to partial observability while the expert has full observability. The proposed approach introduces a regularization term for the actor to encourage the learner to match the action distribution of the expert, which observes the full state. The strength of this term is controlled by \alpha, which is updated by dual gradient descent to match some target divergence. The resulting method COSIL is compared to an array of algorithms on a series of environments with designed partial observability.

**Issues:**

Questions:
- Why does the transferred ADV-Off policy perform worse in the real world when the performances are similar in simulation?
- In Fig. 4b, how does COSIL perform with fixed values for \alpha? I’m curious whether we need to adapt the strength of the divergence at all.
- Would training for longer in Minigrid-Memory and Block-Picking lead to better converged performance for COSIL?

**Quality Of The Limitations Section:**

Limitations are addressed clearly

**Reviewer Expertise:**

3: The reviewer is fairly confident that the evaluation is correct

**Robotics Focus:**

Sufficient demonstration on hardware

**Strengths And Weaknesses:**

Strengths: The paper is overall well-written, and the proposed method COSIL is compared to a strong array of algorithms. COSIL performs either comparably to or better than alternative methods, including ADVISOR which is designed with the same objective in mind. A policy learned by COSIL in simulation is also shown to transfer well onto a physical robot.

Weaknesses: As pointed out in the limitations section,
- to compute the divergence term during training, we require the expert actions at every state visited by the learner, and
- the state expert, while optimal under full observability, can be suboptimal under partial observability.

The second weakness is somewhat concerning to me. The setup studied in this work is a hybrid of RL and IL in that both environment rewards and expert action labels are provided at each time-step, which is strictly more informative than the RL setting and the IL setting. With additional information, I would expect an algorithm designed for this setup should converge to optimal performance or perform better than algorithms that use only rewards and only labels. However, as depicted in Fig. 4a, BC performs better than COSIL when the expert is suboptimal under partial observability.

To account for the potential suboptimality of the state expert, the value of \bar{D} needs to be carefully chosen so that we allow the learner to find a better solution than a suboptimal expert. It seems that too-small values can lead to worse success rates (Fig. 4c) and I suspect that the best choice of \bar{D} would depend on the environment and expert.

**Summary Of Recommendation:**

Overall, this paper studies a practical setting and shows improvements on existing algorithms for this setup. The proposed method is intuitive and rigorously evaluated across multiple environments.

=============

Post-rebuttal: I appreciate the additional analysis on sensitivity to \bar{D} and adapting \alpha. It's helped me understand how alternative design choices affect COSIL.

---

> ### Author Response · Authors · 2022-08-19
> **Response to Reviewer a2e5 (Figure Attached)**
>
> We thank the reviewer for recommending accepting our work. Below we address the reviewer's concerns.
>
> > The second weakness is somewhat concerning to me. The setup studied in this work is a hybrid of RL and IL in that both environment rewards and expert action labels are provided at each time-step, which is strictly more informative than the RL setting and the IL setting. With additional information, I would expect an algorithm designed for this setup should converge to optimal performance or perform better than algorithms that use only rewards and only labels. However, as depicted in Fig. 4a, BC performs better than COSIL when the expert is suboptimal under partial observability.
>
> This is expected for problems with a low optimality gap since COSIL becomes more suitable for problems with a large optimality gap. In Fig. 4a, the domain tested is Minigrid-Crossing (see appendix for the description). This is a specific domain with a very small optimality gap: the paths that lead to a goal cell of an optimal partially observable agent (which only observes a local area around it) and a state agent (which observes the whole grid world) are quite similar. In fact, the actions chosen by the fully observable expert are virtually never wrong in the partially observable case, and there is virtually no information-gathering reason for the partially observable agent to avoid them. In such a domain, standard behavior cloning (BC) is the best learning method because there is no reason to avoid mimicking the state expert as much as possible. Any other learning method would be inferior to a BC agent because it could slow the mimicking process.
>
> > To account for the potential suboptimality of the state expert, the value of \bar{D} needs to be carefully chosen so that we allow the learner to find a better solution than a suboptimal expert. It seems that too-small values can lead to worse success rates (Fig. 4c) and I suspect that the best choice of \bar{D} would depend on the environment and expert.
>
> Similar to a SAC agent [1, 2], which is sensitive to the desired entropy target or $\alpha$, our method is similarly sensitive to the choice of $\bar{D}$. However, in Fig. 4c, our method is stable at half the tested range $0.8 \rightarrow 1.1$. In SAC [1], a common heuristic is to set the target entropy to be $\bar H = \log_2(n)$, where $n$ is the expected number of actions that the final policy is allowed to uniformly choose from at each timestep. In future work, it might be possible to develop a similar heuristic for COSIL and $\bar{D}$.
>
> [1] Haarnoja, Tuomas, et al. "Soft actor-critic algorithms and applications." arXiv preprint arXiv:1812.05905 (2018).
>
> [2] Haarnoja, Tuomas, et al. "Soft actor-critic: Off-policy maximum entropy deep reinforcement learning with a stochastic actor." International conference on machine learning. PMLR, 2018.
>
> > Why does the transferred ADV-Off policy perform worse in the real world when the performances are similar in simulation?
>
> Empirically, when transferring the policies, we found that the policy learned by ADV-Off is less robust than ours. However, given the similar performance in simulation, we believe that the performance would be roughly the same when the sample size increases. We have edited the paper to make this point clearer.
>
> > In Fig. 4b, how does COSIL perform with fixed values for \alpha? I’m curious whether we need to adapt the strength of the divergence at all.
>
> We have updated Fig. 4b (attached here) with the results when $\alpha$ are fixed in $\{0, 0.3, 0.6, 0.9\}$. Adapting $\alpha$ seems more performant than fixing $\alpha$.
>
> > Would training for longer in Minigrid-Memory and Block-Picking lead to better converged performance for COSIL?
>
> We have doubled the training time to show that COSIL can converge. For now, we only include the performance of COSIL in the two domains, as requested. We __attached__ the figures here and also updated the paper (appendix H.3, Fig. 16). We will run other baselines and update the result later.

---

> ### Author Response · Authors · 2022-08-25
> **Last two days of rebuttal discussion**
>
> Dear Reviewer,
>
> Could you take a look at our rebuttal to see whether it addressed your concerns? There are only 2 days left and we are concerned not hearing from you. Thanks!

---

### Official Review · Reviewer_Spnc · 2022-07-31

**Originality:** Good
**Technical Quality:** Good
**Clarity Of Presentation:** Poor
**Impact:** 2

**Recommendation:**

Weak Reject: I recommend rejecting the paper, but will not argue for my recommendation if the majority of other reviewers have a different opinion.

**Summary:**

The paper is interested in reinforcement learning in partially observed tasks. Specifically, the approach aims to make use of an online expert with full observability to improve the training of an SAC agent. The key insight is that the fully observable experts actions may not always be optimal for the partially observable agent.

The methods two main components are:
- Adding a BC term to the SAC reward.
- Dynamically adjusting the weight of the BC term to map to a target BC term.

Experiments suggest that the method outperforms SAC+BC, DAgger, and other relevant baselines on some simulated and one real robot task.

**Issues:**

See Strengths and Weaknesses.

**Quality Of The Limitations Section:**

Limitations are addressed clearly

**Reviewer Expertise:**

3: The reviewer is fairly confident that the evaluation is correct

**Robotics Focus:**

Sufficient demonstration on hardware

**Strengths And Weaknesses:**

*Strengths*

- Studying training RL policies under partial observability is an interesting and relevant problem for robotic learning.
- The motivation makes sense in partially observable tasks, a fully observable expert actions may not always be optimal for the partially observed agent.
- The paper has experiments on a physical robot.

*Weaknesses*

First, I have some questions about the method:
- The proposed method as I understand it simply amounts to adding a BC loss to the reward with an online queryable expert at each state. Section 4 frames this as a KL divergence, but looking in the appendix the online expert is deterministic and this is simply amounts to an MSE loss. This added BC objective is commonplace in many IL+RL algorithms algorithms, the paper could do a much better job at explaining how their instantiation is different from prior work.
- In general, I don't understand what about the method is "Soft" Imitation Learning. The introduction and Section 4 emphasizes "soft" but what does this mean and how does it describe the method? I found the paragraph in lines 145-158 unclear in explaining this.
- I also don't understand fully why the weight on the BC term is adapted to hit some target divergence? I understand that the method is trying to balance maximizing reward vs. matching the potentially suboptimal expert, but I would expect the choice of alpha to do this. It would be good to see an ablation of the COSIL method with the adaptive re-weighting part removed.

Second, I have some questions about the experiments:
- What is different between the proposed method and the SAC+BC baseline? Is it just that in COSIL the BC loss is part of the reward? I find it surprising that performs so much worse than COSIL considering how similar they seem.
- I would imagine that if the expert is suboptimal as described, the pure SAC agent would eventually outperform COSIL. It would be interesting to see at what point SAC starts to outperform.

Third, overall the paper clarity could be improved significantly. The paper could better explain what about the current method is novel and why it is important for partially observed tasks. It should also have more details about the method implementation in the main text, including the algorithm block which is currently in the appendix.

I also have some larger questions about the practicality of the problem setup. For example, the assumption that there is an online expert who gives action labels not just on current online states, but any state the agent wants to query (e.g. for target states)  is strong. For example a human teleoperator cannot do this in the real world, which seems to limit the approach largely to simulation. And if the policies are learning in simulation, then sample efficiency is less of a concern, and pure SAC which will likely eventually reach higher asymptotic performance would be a better choice.

**Summary Of Recommendation:**

Overall the paper has some solid insights around learning in partial observability and using a state based expert. But the paper currently does not clearly explain and motivate the method, and I have some open questions about the method, experiments, and general setup that hopefully can be addressed in the rebuttal.

---

> ### Author Response · Authors · 2022-08-19
> **Response to Reviewer Spnc (Figure Attached) (1/3)**
>
> Authors thank the reviewer for useful feedback. Below we address the reviewer's concerns.
>
> ---
>
> > The proposed method as I understand it simply amounts to adding a BC loss to the reward with an online queryable expert at each state. Section 4 frames this as a KL divergence, but looking in the appendix the online expert is deterministic and this is simply amounts to an MSE loss. This added BC objective is commonplace in many IL+RL algorithms algorithms, the paper could do a much better job at explaining how their instantiation is different from prior work.
>
> In section 4, we try to be as generic as possible and only claim that $D$ is some notion of divergence between two distributions.  The KL divergence is only one possible example, while the MSE could be another.  We expect that a good choice of divergence might depend on the environment and the nature of the action-space, and in practice, we use the MSE divergence simply because ADVISOR similarly employs the MSE divergence in their experiments (despite the respective theory being described in terms of KL).
>
> For the difference between our method and the common IL+RL algorithm, please see our response to the reviewer's question later.
>
> ---
>
> > In general, I don't understand what about the method is "Soft" Imitation Learning. The introduction and Section 4 emphasizes "soft" but what does this mean and how does it describe the method? I found the paragraph in lines 145-158 unclear in explaining this.
>
> We chose to use the term "soft" to describe our method for two reasons: first, it's because we are combining the RL objective and the IL objective, so this is not just 100% imitation learning.  Second, most importantly, was to represent the similarity with SAC, which combines the RL objective and a max-entropy objective, but does so specifically by treating the max-entropy objective as a pseudo-reward.  Since our method applies a similar structure, but for imitation learning, we think that "soft" imitation learning is an adequate name.
>
> ---
>
> > I also don't understand fully why the weight on the BC term is adapted to hit some target divergence? I understand that the method is trying to balance maximizing reward vs. matching the potentially suboptimal expert, but I would expect the choice of alpha to do this. It would be good to see an ablation of the COSIL method with the adaptive re-weighting part removed.
>
> The reviewer is correct that $\alpha$ directly represents how to combine and balance the RL objective and the IL objective.  However, tweaking $\alpha$ directly is hard because of two issues: the first is that the semantics of $\alpha$ is hard to understand and requires being able to balance return values associated with the task with the divergence values.  The second is that the correct value of $\alpha$ might vary and change throughout the training, so a constant $\alpha$ might not suffice.  Similar issues have been known to exist for SAC, which is the main inspiration for COSIL.
>
> Therefore, we employ an auto-tuning technique that is similar to one that was developed in the latest version of SAC [1]  to improve over a previous version [2] which fixed $\alpha$ before training.  The technique adaptively changes the SAC $\alpha$ so that the entropy of the desired policy match with a pre-chosen value. The main motivation for the change is that the correct value of $\alpha$ is difficult to tune, while the correct value of the target entropy is easier to choose.
>
> In our setting, the target divergence indicates how big the difference between the behavior of an optimal policy under partial observability and the state expert. Empirically, we found that automatically adapting $\alpha$ based on the target divergence was easier than
> setting $\alpha$ for each environment.
>
> We also incorporate an ablation of Car-Flag with the adaptive re-weighting part removed. Please see the updated Fig. 4 (__attached__). Based on the figure, an adaptive $\alpha$ that changes based on some target divergence seems to be more efficient than setting it to a fixed value or setting it based on some predetermined decaying schedule.
>
> __References__:
>
> [1] Haarnoja, Tuomas, et al. "Soft actor-critic algorithms and applications." arXiv preprint arXiv:1812.05905 (2018).
>
> [2] Haarnoja, Tuomas, et al. "Soft actor-critic: Off-policy maximum entropy deep reinforcement learning with a stochastic actor." International conference on machine learning. PMLR, 2018.

---

> > ### Author Response · Authors · 2022-08-19
> > **Continue... (2/3)**
> >
> > > What is different between the proposed method and the SAC+BC baseline? Is it just that in COSIL the BC loss is part of the reward? I find it surprising that performs so much worse than COSIL considering how similar they seem.
> >
> > The difference between our method and the SAC+BC baseline is that the BC loss is part of the reward.  While this may seem like a minor detail at first, it carries significant theoretical and practical implications.  The relationship between COSIL and SAC+BC is analogous to that that has been established between SAC (which has an entropy term as an additional reward) and A2C with entropy regularization (which has an entropy term that is not incorporated as an additional reward), where SAC has also been established as the better choice.
> >
> > In our setting, incorporating the BC objective as a pseudo-reward can directly guide the policy to seek situations where it can act as if it were fully observable, i.e., indirectly, this can help learn information-gathering behaviors.
> >
> > We would also want to point out that a fixed static combination between BC and SAC can be suboptimal. The reason is that the optimal behavior under partial observability can sometimes be similar but sometimes can be vastly different from the actions of the fully observable expert (state expert). For instance, when the agent is certain about the state, it should mimic the state expert. However, before that, it must act differently from the state expert because it needs to gather more information about the state (the state expert hardly has to do this because it knows the state). This explains the learning curve of BC+SAC in Fig. 3, where it can perform well in LunarLander-P (a domain with a small optimality gap, i.e., a small level of partial observability as only a short history is needed to infer the missing information) but generally underperforms in other domains. Having a fixed weight between the BC loss and the SAC loss will prevent the agent from adaptively changing the behavior as needed because the same fixed weights are assigned for the BC and RL losses. In fact, ADVISOR is an adaptively weighted version of BC+PPO in which the weight can be changed adaptively per state (please see the appendix for the description of ADVISOR).
> >
> > ---
> >
> > > I would imagine that if the expert is suboptimal as described, the pure SAC agent would eventually outperform COSIL. It would be interesting to see at what point SAC starts to outperform.
> >
> > With a state expert that is extremely suboptimal under partial observability, there are going to be a few ways to use it to help learn a good partially observable agent, with the main exception being its ability to show interesting parts of the state space.  Fig. 6 demonstrates that both COSIL and ADVISOR are outperformed by a pure (memory-based) SAC agent. Nevertheless, we have an ablation study that compares a pure SAC agent and COSIL when the expert is suboptimal in a modified version of Bumps-2D (as shown in Fig. 6). SAC starts to outperform at about half of the training (at about 50K environment steps). However, we think the comparable performance is highly dependent on the domain.
> >
> > ---
> >
> > > Third, overall the paper clarity could be improved significantly. The paper could better explain what about the current method is novel and why it is important for partially observed tasks. It should also have more details about the method implementation in the main text, including the algorithm block which is currently in the appendix.
> >
> > We have rewritten parts of the paper to improve the clarity of the sections indicated by all reviewers.

---

> > > ### Author Response · Authors · 2022-08-19
> > > **Continue... (3/3)**
> > >
> > > > I also have some larger questions about the practicality of the problem setup. For example, the assumption that there is an online expert who gives action labels not just on current online states, but any state the agent wants to query (e.g. for target states) is strong. For example a human teleoperator cannot do this in the real world, which seems to limit the approach largely to simulation. And if the policies are learning in simulation, then sample efficiency is less of a concern, and pure SAC which will likely eventually reach higher asymptotic performance would be a better choice.
> > >
> > > COSIL operates in an asymmetric learning setting, where privileged information (e.g., the state and the state expert) is available during training (e.g., in simulation) but not during execution. The privileged information is exploited to learn a partially observable policy, which can be later deployed without needing to access the privileged information. Therefore, the resultant policy can be deployed or transferred to the real robot in the real world -- which is demonstrated in our robot experiment in 5.6. This learning paradigm has been successful in previous work, and it fits the current robot learning paradigm: learning policies in simulation and deploying them in the real world (more about this, please see our response to a similar point by the reviewer **6hqs**).
> > >
> > > For tasks with an exploration bottleneck, pure SAC would not be able to solve them efficiently even if it has access to the state or a state expert (we must know a sufficient history statistic to act optimally under partial observability). Take the Block-Picking as an example. A memory-based SAC agent will take a long time to learn to pick a block, examine if the block is movable, and select the other block if it discovers it cannot move the block. In fact, Fig. 3 shows that SAC completely failed in this task. Block-Picking is a very challenging task because the task has a continuous action space, the horizon can be long, the agent has to explore efficiently and actively, and the environment only provides a sparse reward. For this task, COSIL guided by the state expert, can solve the task, and the policy learned in simulation can transfer to the real robot, as shown in 5.6.

---

> ### Author Response · Authors · 2022-08-25
> **Last two days of rebuttal discussion**
>
> Dear Reviewer,
>
> Could you take a look at our rebuttal to see whether it addressed your concerns? There are only 2 days left and we are concerned not hearing from you. Thanks!

---

### Official Review · Reviewer_oqag · 2022-08-01

**Originality:** Good
**Technical Quality:** Fair
**Clarity Of Presentation:** Excellent
**Impact:** 3

**Recommendation:**

Weak Reject: I recommend rejecting the paper, but will not argue for my recommendation if the majority of other reviewers have a different opinion.

**Summary:**

This paper proposes Cross-Observability Soft Imitation Learning (COSIL), a method for augmenting partially-observable RL training with a fully-observable RL policy to improve performance during training. The method introduces the COSIL objective, which is similar to the SAC critic but swaps out the entropy term for a distance metric measuring the similarity between the partially-observable policy and fully-observable policy. The paper studies experimental settings with varying amounts of optimality gap, which consists of the behavior policy performance difference between expert partially-observable policies and fully-observable policies. Specifically, they study 6 different environments that include some form of partial observability, such as withholding geometric or physical properties of the scene.

**Issues:**

- Were the fully observable policies trained in parallel or pretrained? In section 2, it’s claimed that a drawback of ADVISOR and A2D are that the state expert must be pretrained, but it’s not clear which setting the experimental settings use. Are there experiments comparing using a fixed pretrained state expert vs. a state expert that is trained in parallel with the partially observable expert?
- Were the BC baselines in 5.2 fully observable or partially observable experts?
- The baseline for ADV-OFF seems poor in the most complex domain, block picking. The variance is high, and there is a strange dip early in training. Can you run more seeds?


**Quality Of The Limitations Section:**

Limitations are addressed clearly

**Reviewer Expertise:**

4: The reviewer is confident but not absolutely certain that the evaluation is correct

**Robotics Focus:**

Highly relevant to robotics but no hardware experiments

**Strengths And Weaknesses:**

Strengths:
- The paper is easy to follow and well written
- The motivation and comparison with SAC is helpful

Weaknesses:
- The significance is limited because the empirical settings seem quite contrived (ie. adding features to the environment that are not standard in robot learning). It feels that each of the experimental settings introduce partial observability in fairly forced ways by adding specific MDP adjustments to normal robot learning tasks. More relevant experiments can be motivated by similar works in multi-modal learning that consider state masking or modality masking.
- Similar to the above point, it’s unclear how widely applicable COSIL is or whether it will only work in specific partially observable MDPs; with a very suboptimal state expert, performance can actually be worse, as shown in Figure 6. This may require very careful selection of when and how to apply COSIL.
- The performance of COSIL in low optimality gap environments is not great
- The real world performance improvement is hard to gauge, perhaps due to the small evaluation sample size. The evaluation also introduced a large sim-to-real gap as mentioned in 5.6.


**Summary Of Recommendation:**

The motivation with SAC is elegant and helps introduce an original and interesting method. My concern is that the impact is limited by the empirical results, which are mediocre even on fairly contrived settings. This seems to suggest challenges for the applicability and significance of the method, even if it is sound and novel.

---

> ### Author Response · Authors · 2022-08-19
> **Response to Reviewer oqag (1/3)**
>
> We thank the reviewer for the useful feedback. Below we address the reviewer's concerns.
>
> ---
>
> > The significance is limited because the empirical settings seem quite contrived (i.e. adding features to the environment that are not standard in robot learning). It feels that each of the experimental settings introduce partial observability in fairly forced ways by adding specific MDP adjustments to normal robot learning tasks. More relevant experiments can be motivated by similar works in multi-modal learning that consider state masking or modality masking.
>
> We would argue that our domains represent a diverse set of practical partial observability, highly related to the reviewer's suggestions. Bumps-1D and Bumps-2D can represent the tasks of goal-reaching navigation with tactile feedback as an additional modality. It is indeed our original thought when creating the domain. However, we substitute the tactile data with the finger's angle, which is easier to simulate than simulating force and tactile sensing in a simulator like MuJoCo [1].  Minigrid-Memory is a standard POMDP task in the Minigrid suite [2], which represents the practical setting when the agent can only observe a local view around it. Car-Flag can represent a practical exploration task in an unknown environment as the agent must actively navigate to interesting regions for information. Lunar-Lander domains use state masking, as the reviewer has suggested. Similar to Bumps-1D and Bumps-2D, Block-Picking can have tactile data involved besides depth images to examine whether a block is pickable or not, but we observe that using a depth image can provide the same information; therefore, we use depth images instead. As a final thought, compared to typical POMDP domains commonly used in the literature, such as continuous control with masked states [3], noisy observations [4], or flickering Atari games [4, 5] with some probability, our domains require long-term memory and active information gathering. In our view, they are more challenging and practical.
>
> __References__:
>
> [1] Todorov, Emanuel, Tom Erez, and Yuval Tassa. "Mujoco: A physics engine for model-based control." 2012 IEEE/RSJ international conference on intelligent robots and systems. IEEE, 2012.
>
> [2] Chevalier-Boisvert, Maxime, Lucas Willems, and Suman Pal. "Minimalistic gridworld environment for openai gym." (2018).
>
> [3] Ni, Tianwei, et al. "Recurrent model-free rl can be a strong baseline for many pomdps." International Conference on Machine Learning. PMLR, 2022.
>
> [4] Igl, Maximilian, et al. "Deep variational reinforcement learning for POMDPs." International Conference on Machine Learning. PMLR, 2018.
>
> [5] Hausknecht, Matthew, and Peter Stone. "Deep recurrent q-learning for partially observable mdps." 2015 aaai fall symposium series. 2015.
>
> ---
>
> > Similar to the above point, it’s unclear how widely applicable COSIL is or whether it will only work in specific partially observable MDPs; with a very suboptimal state expert, performance can actually be worse, as shown in Figure 6. This may require very careful selection of when and how to apply COSIL.
>
> We agree that the performance of COSIL would depend on the optimality of the state expert under partial observability.  With a state expert that is extremely suboptimal under partial observability, there are going to be few ways to use it to help learn a good partially observable agent, with the main exception being the its ability to show interesting parts of the state space. COSIL tries to exploit the fully observable expert in both ways:  both when the expert provides some trajectories that are successful under partial observability, and when it can guide the history agent to explore by showing interesting parts of the state space. For more about the setting where COSIL is applicable, please see our response to the reviewer **6hqs**.

---

> > ### Author Response · Authors · 2022-08-19
> > **Continue..., (Figure Attached) (2/3)**
> >
> > > The performance of COSIL in low optimality gap environments is not great
> >
> > This is to be expected, and a method like COSIL would show its advantages in environments with a large optimality gap, i.e., with a larger amount of partial observability. In domains with a small optimality gap, the desired partially observable behavior is similar to the state expert's behavior, i.e., the environment has low amounts of partial observability, and the best method would be standard behavior cloning (BC) because it is optimally sufficient to mimic the actions of the state expert. Any other learning methods would likely be inferior to BC.
> >
> > ---
> >
> > > The real world performance improvement is hard to gauge, perhaps due to the small evaluation sample size. The evaluation also introduced a large sim-to-real gap as mentioned in 5.6.
> >
> > Our real-world experiment aims to show that COSIL can help train a performant policy in a challenging partially observable task. Empirically, it seems more robust than ADVISOR, with higher real-world success rates (see Table 1). However, we would expect they perform somewhat similarly with bigger sample size, given their similar performance in simulation. We have edited the paper to make this point clearer. The large sim-to-real gap is a practical aspect which goes beyond the scope of our work, but which we overcame by adding noise to the depth images during training.
> >
> > ---
> >
> > > Were the fully observable policies trained in parallel or pretrained? In section 2, it’s claimed that a drawback of ADVISOR and A2D are that the state expert must be pretrained, but it’s not clear which setting the experimental settings use. Are there experiments comparing using a fixed pretrained state expert vs. a state expert that is trained in parallel with the partially observable expert?
> >
> > We agree we should be more clear in explaining this aspect. __Our experiments assume that a fully observable state expert is given for all baselines before training__. We have edited the related work and the experiment section to clarify our points. In fact, both ADVISOR and A2D **jointly** train an additional actor (ADVISOR) and a state expert (A2D) and the partially observable policy. The additional actor in ADVISOR takes in the history but will be trained to mimic the given state expert (see the description of ADVISOR in the appendix). In contrast, A2D jointly trains (from scratch) a student policy with a state expert, which is required to be updateable through gradient updates. For this reason, both ADVISOR and A2D can only work well when the additional actor/state expert is properly trained, i.e., behaving less randomly. This is a clear advantage of our approach. Finally, we did not include any experiments where the state expert is trained jointly with the partially observable policy because only A2D would be a fair baseline in such a setting.
> >
> > > Were the BC baselines in 5.2 fully observable or partially observable experts?
> >
> > All experts are assumed to be fully observable. If we have a partially observable expert, then we can use standard behavior cloning with no need for ADVISOR or COSIL. However, obtaining such an expert is still difficult. Even if the state is available, getting a partially observable expert would require some sufficient statistics of the history ($h$), like the belief states $p(s|h)$. Computing the belief states impractically requires the dynamics of the environment, which most classical POMDP solution methods like SARSOP [1], DESPOT [2] assume to have, while modern model-free methods avoid. In contrast, by having the states during training, we can get the state expert easier, e.g., by running the standard RL method for MDPs. This observation has inspired our work.
> >
> > [1] Kurniawati, Hanna, David Hsu, and Wee Sun Lee. "Sarsop: Efficient point-based pomdp planning by approximating optimally reachable belief spaces." Robotics: Science and systems. Vol. 2008. 2008.
> >
> > [2] Somani, Adhiraj, et al. "DESPOT: Online POMDP planning with regularization." Advances in neural information processing systems 26 (2013).
> >
> > > The baseline for ADV-OFF seems poor in the most complex domain, block picking. The variance is high, and there is a strange dip early in training. Can you run more seeds?
> >
> > We have doubled the number of seeds (4 -> 8 seeds) for both ADV-OFF and COSIL. Please see the updated Fig. 3 __attached__.
> >
> > __Clarifying Question__
> >
> > Could the reviewer explain this comment, "Robotics Focus: Highly relevant to robotics but no hardware experiments"? We have successfully transferred a policy to a real robot.

---

> > > ### Author Response · Authors · 2022-08-27
> > > **Continue... (3/3)**
> > >
> > > > My concern is that the impact is limited by the empirical results, which are mediocre even on fairly contrived settings
> > >
> > > We have addressed the practical representation of our domains. Next, allow us to point out that COSIL is the best agent in the seven domains tested, as shown in Fig. 3. It did not outperform significantly in domains with a small optimality gap, i.e., the level of partial observability is small, such as LunarLander domains because these domains only require short memory (few recent observations are sufficient to infer the missing components of the state). Similarly, in Minigrid-Crossing (Fig. 4a), it (and ADVISOR as well) is outperformed by behavior cloning because this domain has an insignificant optimality gap. Please see our response to the reviewer __a2e5__ below for more explanation.

---

> ### Author Response · Authors · 2022-08-25
> **Last two days of rebuttal discussion**
>
> Dear Reviewer,
>
> Could you take a look at our rebuttal to see whether it addressed your concerns? There are only 2 days left and we are concerned not hearing from you. Thanks!

---

### Official Review · Reviewer_6hqs · 2022-08-01

**Originality:** Good
**Technical Quality:** Very Good
**Clarity Of Presentation:** Very Good
**Impact:** 2

**Recommendation:**

Weak Reject: I recommend rejecting the paper, but will not argue for my recommendation if the majority of other reviewers have a different opinion.

**Summary:**

- goal: RL for partial observability
- idea: leverage good fully observable policies (state experts) to help train partially observable policies
- introduced objective: minimizing divergence (e.g. KL) between expert and partially observable agent
- training in such a way is suboptimal, but applying objective in a soft fashion is beneficial
- car-flag: agent learns to accelerate only once it gains more information
- lander: policy less aggressive than expert due to partial observability
- block-picking: agent doesn’t see color. quickly moves to other block once it sees it cannot pick it. real setup: use sim policies with real depth maps.
- Results:
  - does well across the board compared to dagger and RL methods
  - method performs well in both optimality gap and no gap.
- limitation: requires constant access to expert actions at every state during training. proposes fix by learning from limited demonstrations.


**Issues:**

- maybe the authors can motivate better / clarify real life scenarios where the approach is directly applicable? (where a state expert is available)
- The video and voice over is too fast, it’s hard to follow


**Quality Of The Limitations Section:**

Limitations are addressed clearly

**Reviewer Expertise:**

3: The reviewer is fairly confident that the evaluation is correct

**Robotics Focus:**

Sufficient demonstration on hardware

**Strengths And Weaknesses:**

strengths:
- paper is pretty well explained
- policy can handle partial observability.
- it seems that the policies are doing as well as the expert but in a more careful manner until they observe the right information because they aren’t as confident as the state expert, this seems like a good thing.
- good results across all tasks, whether there is optimality gap or not

weaknesses:
- it’s not very clear to me in what real world scenario you have access to a state expert?
- not clear how practical or useful this is in the real world
- real setup is pretty limited


**Summary Of Recommendation:**

Well written paper, good results on narrow setups, but rather limited real setup, and not clear how practical or useful this is in the real world.

---

> ### Author Response · Authors · 2022-08-19
> **Response to Reviewer 6hqs (New Video Attached)**
>
> **Comment:**
>
> We thank the reviewer for the useful feedback. Below we address the reviewer's concerns.
>
> > it’s not very clear to me in what real world scenario you have access to a state expert? not clear how practical or useful this is in the real world. maybe the authors can motivate better / clarify real life scenarios where the approach is directly applicable? (where a state expert is available)"
>
> COSIL operates in the asymmetric learning setting, a variant of offline RL in which privileged information (e.g., state, which can be used to get a state expert) is available during training (e.g., using high-fidelity simulators) but not during execution. This privileged information is used to accelerate learning but is not used after the training is performed when the agent begins to operate in the real world where this information is not available.
>
> Previous work has adopted this asymmetric learning paradigm [1:7] for both MDPs and POMDPs with good results. It fits the common theme of a robot that learns in simulators and then transfers the learned policies to the real world. In a multi-agent setting, the popular centralized-training-decentralized-execution (CTDE) is another variant of this paradigm, where other forms of privileged information (all joint observations [8], all histories [9], or the state [10]) are accessible only during the offline training.
>
> We have edited the introduction and the method in the paper to motivate better; please see the new version of the paper __in the summary comment on top__.
>
> __References__:
>
> [1] Baisero, Andrea, et al. "Asymmetric DQN for Partially Observable Reinforcement Learning." The 38th Conference on Uncertainty in Artificial Intelligence. 2022.
>
> [2] Pinto, Lerrel, et al. "Asymmetric actor critic for image-based robot learning." Robotics-Science and Systems XIV, 2018.
>
> [3] Warrington, Andrew, et al. "Robust asymmetric learning in pomdps." International Conference on Machine Learning. PMLR, 2021.
>
> [4] Chen, Dian, et al. "Learning by cheating." Conference on Robot Learning. PMLR, 2020.
>
> [5] Nguyen, Hai, et al. "Belief-grounded networks for accelerated robot learning under partial observability." Conference on Robot Learning. PMLR, 2020.
>
> [6] Baisero, Andrea, et al. "Unbiased Asymmetric Reinforcement Learning under Partial Observability." Proceedings of the 21st International Conference on Autonomous Agents and Multiagent Systems. 2022.
>
> [7] Wilson, Matthew, and Tucker Hermans. "Learning to manipulate object collections using grounded state representations." Conference on Robot Learning. PMLR, 2020.
>
> [8] Foerster, Jakob, et al. "Counterfactual multi-agent policy gradients." Proceedings of the AAAI conference on artificial intelligence. Vol. 32. No. 1. 2018.
>
> [9] Wang, Rose E., Michael Everett, and Jonathan P. How. "R-MADDPG for partially observable environments and limited communication." arXiv preprint arXiv:2002.06684 (2020).
>
> [10] Yang, Jiachen, et al. "Cm3: Cooperative multi-goal multi-stage multi-agent reinforcement learning." arXiv preprint arXiv:1809.05188 (2018).
>
> ---
>
> > real setup is pretty limited
>
> Our real robot experiment is a very challenging, partially observable manipulation task. The reasons are that the exploration is hard, the reward is sparse, the action space is continuous, the agent is required to perform active information gathering, and the task can be long-horizon. If learning from scratch, a memory-based agent would have a lot of difficulties learning how to pick a block, examine if the block is pickable, and then move to pick the other block if the earlier block is unpickable. With the guided exploration from the state expert, which can always pick the movable block in simulation, COSIL can learn a policy purely in simulation for this task that also works on hardware after being transferred.
>
> ---
>
> > The video and voice over is too fast, it's hard to follow
>
> We have fixed these issues in the new version of the video. We lengthened the parts showing the policies for Bumps-1D and -2D, which we assume is what made it difficult to follow. However, we want to keep the video under 3 minutes to respect the reviewers' time. We also __attach__ the video here in case the reviewer wants to take a look.
>
> **Zip File:**
>
> /attachment/b78246a37bb1020e59b5918759e53a282f3caa84.zip

---

> ### Author Response · Authors · 2022-08-25
> **Last two days of rebuttal discussion**
>
> Dear Reviewer,
>
> Could you take a look at our rebuttal to see whether it addressed your concerns? There are only 2 days left and we are concerned not hearing from you. Thanks!

---

### Author Response · Authors · 2022-08-19
**Summary of Changes Made (New Paper + Supplementary - Attached)**

**Comment:**

Below we summarize the changes to the paper and the supplementary materials after incorporating all the comments made by the reviewers:

- Rewrite parts of the paper to clarify the practical setting of our method and how different our method to a normal RL+IL approach
- Run an ablation study of not adapting $\alpha$ but instead fixing $\alpha$ at different values
- Train longer for COSIL in Minigrid-Memory and Block-Picking to check the convergence, resulting in a new figure (Fig. 16) in Appendix H.3
- Increase the number of seeds for COSIL and ADVISOR in Block-Picking and update Fig. 3
- Update the supplementary video to make it easier to follow

__We also attached the new version of the paper and the supplementary here__. We highlight the changes that we made in blue.

**Zip File:**

/attachment/eb8fb39eced42c153a9e49bd32f2cab2b10581a3.zip

---

### Meta-Review · Area_Chair_oMPb · 2022-08-08

**Recommendation:** Accept (Poster)
**Confidence:** 3

**Metareview:**

This paper initially received (weak accept, 3X weak reject).
During the discussion, an agreement was not reached, and the current scores are (weak accept, borderline accept, 2X weak reject). I base my decision on the reviews and discussion, and on my own review of the paper.

The main point of conflict between reviewers is the limitations of the asymmetric learning setup - having access to a fully observable policy during training, but learning a partially observable policy for test time.

As several reviewers pointed out, this setup is indeed limiting in many realistic scenarios, and the experiments in the paper therefore seem contrived.

However, POMDPs are a very difficult problem, and this paper proposes a very simple and novel approach. It may be that future work relaxes some of the assumptions (in particular, using sim2real, or in a meta-RL setting where some tasks are known), and exposing this work to the CoRL community may foster such follow-ups.

Strengths:
- Important, difficult problem
- Good empirical results (relatively extensive in sim, limited on real)

Weaknesses:
- Real world practicality not clear enough
- Some important parameters in the method are environment dependent and may be hard to choose

**Best Paper Nomination:**

No

---

> ### Author Response · Authors · 2022-08-19
> **Response to Area Chair oMPb  (Figure Attached)**
>
> We thank the area chair for thoughtful comments; below we address the area chair's concerns:
>
> > Real world practicality not clear enough
>
> We have addressed similar points made by other reviewers below. To sum up, our method works in an asymmetric learning setting when there is privileged information (state and the expert state) during training (e.g., from simulators) but not during online execution. Our method then leverages the state expert to learn a partially observable policy that can be transferred to the real world (we demonstrated such transfer in our robot experiment in 5.6) without needing to access the privileged information anymore. This learning paradigm has been used widely and successfully in previous work (see below for citations). It also fits with the current theme of robot learning, which learns in simulators (which may provide privileged information cheaply) and transfers the learned policy to the real world.
>
> We have modified the paper so that the real-world practicality is made clearer.
>
> > Some important parameters in the method are environment dependent and may be hard to choose
>
> The reviewers have concerns about the sensitivity of choosing the target divergence $\bar{D}$ for each environment and the necessity of adapting $\alpha$. In our ablation study in Fig. 4c, we have pointed out that our method is stable during half of the tested range for $\bar{D}$. Moreover, we performed an additional ablation study (see the updated Fig. 4b - __attached__) to show that adapting $\alpha$ can be more beneficial than fixing it. Similar to SAC, there is not a hard requirement that $\alpha$ must be adaptively changed based on $\bar{D}$. However, empirically, we found doing that is easier than manually tuning $\alpha$ for each environment, and it has satisfactory performance in our domains.